# Hidden Entity Detection from GitHub Leveraging Large Language Models

Lu Gan[1,2,*,‡], Martin Blum[3,‡], Danilo Dessí[1], Brigitte Mathiak[1], Ralf Schenkel[3] and Stefan Dietze[1,2]

[1]*GESIS – Leibniz Institute for the Social Sciences, Köln, Germany*

[2]*Heinrich Heine University Düsseldorf, Germany*

[3]*University of Trier, Trier, Germany*

## Abstract

Named entity recognition is an important task when constructing knowledge bases from unstructured data sources. Whereas entity detection methods mostly rely on extensive training data, Large Language Models (LLMs) have paved the way towards approaches that rely on zero-shot learning (ZSL) or few-shot learning (FSL) by taking advantage of the capabilities LLMs acquired during pretraining. Specifically, in very specialized scenarios where large-scale training data is not available, ZSL / FSL opens new opportunities. This paper follows this recent trend and investigates the potential of leveraging Large Language Models (LLMs) in such scenarios to automatically detect datasets and software within textual content from GitHub repositories. While existing methods focused solely on named entities, this study aims to broaden the scope by incorporating resources such as repositories and online hubs where entities are also represented by URLs. The study explores different FSL prompt learning approaches to enhance the LLMs' ability to identify datasets and software mentions within repository texts. Through analyses of LLM effectiveness and learning strategies, this paper offers insights into the potential of advanced language models for automated entity detection.

### Keywords
Named Entity Recognition, Large Language Model, Knowledge Graph

## 1. Introduction & Background

Knowledge graph (KG) construction is a task performed in various domains to organize and structure entities and their relationships. This is performed mainly either manually [1] or automatically via supervised or semi-supervised pipelines [2, 3, 1]. Typically, knowledge graph population includes several tasks such as Named Entity Recognition (NER) [4], relation extraction [5], event extraction [6], and link prediction [7]. Earlier, these tasks were tackled using supervised approaches facilitated by a large amount of domain-specific labeled data. Recently, with the proliferation of Large Language Models (LLM), the research community

*Workshop on Deep Learning and Large Language Models for Knowledge Graphs 2024, (DL4KG'24), ACM KDD'24, Barcelona, Spain.*

*Corresponding author.

†These authors contributed equally.

✉ lu.gan@gesis.org (L. Gan); blumma@uni-trier.de (M. Blum); danilo.dessi@gesis.org (D. Dessí); brigitte.mathiak (B. Mathiak); schenkel@uni-trier.de (R. Schenkel); stefan.dietze@gesis.org (S. Dietze)

0000-0001-5844-3021 (L. Gan); 0000-0003-0005-6162 (M. Blum); 0000-0003-3843-3285 (D. Dessí); 0000-0003-1793-9615 (B. Mathiak); 0000-0001-5379-5191 (R. Schenkel); 0009-0001-4364-9243 (S. Dietze)

has observed unprecedented advancements in the Natural Language Processing (NLP) field. However, the exploration for the KG construction task has not been fully explored due to the huge diversity of domains and usable inputs as sources [8]. In fact, despite the recent advancements, significant limitations persist, especially when it comes to handling complex concepts unique to specialized domains [3]. Furthermore, whereas entity detection methods mostly rely on extensive training data, LLMs have paved the way for approaches that utilize zero-shot learning (ZSL) or few-shot learning (FSL) by taking advantage of the capabilities LLMs acquired during pretraining. Thus, in very specialized scenarios where large-scale training data is not available, ZSL / FSL opens new opportunities. This is especially prominent in scientific research where a deep understanding of particular entities and their interconnections is required, while no sufficient training data exists.

Existing works build KGs either manually or by adopting supervised pipelines on scientific literature [9, 10], ignoring resources that are released along with or cited by research papers, such as source code repositories, datasets, or Machine Learning (ML) models. For example, ML models are available on hubs such as Huggingface [11] and PyTorch Hub [12], software source code is found on hosting services such as GitHub [13] and BitBucket [14], and datasets are provided in repositories such as Zenodo [15]. However, other places exist where code and datasets can be stored, e.g., they can be released as downloads from servers of institutions or organizations and, specifically to datasets, they can be linked to repositories where an associated source code is hosted. This lack of standardization in releasing software and datasets is an issue in making research reproducible and reusable, thus contributing to the state of the art's crisis [16]. Focusing on where datasets are shared raises two main issues: i) datasets shared in repositories are not easily findable and their reuse is limited, and ii) it is not easy to track state-of-the-art results on these datasets since they are not referenced uniformly (i.e., they do not have an associated Digital Object Identifier (DOI) and citations are often embedded in continuous text or footnotes instead of bibliographies).

To address these issues, in this paper, we present our efforts to automatically discover datasets and software hidden on README pages in GitHub repositories using Large Language Models, and describe our experience in extracting them for a later knowledge graph population. This task diverges from traditional NER approaches for three main reasons: first, comprehending the content linked to a URL solely from its context poses significant difficulty; second, URLs are not frequent tokens encountered in vocabulary, leading language models to potentially lack sufficient information for accurate interpretation; third, it is not trivial to retrieve sufficient contexts to infer the intention of URLs from the structured README pages. Specifically, our study examines the effectiveness of LLMs in two aspects: first, their capability to identify software and datasets represented by URLs within GitHub repository READMEs, and second, their performance in classifying URLs found in these repositories. To do so, we present our analysis in FSL settings due to the lack of available training data. In summary, the contributions of this paper are: i) an analysis of two LLMs and their quantized models to detect datasets and software in text from GitHub repositories, ii) an analysis of few-shot prompts to teach an LLM to detect these mentions, and iii) a manually annotated dataset of 811 GitHub repositories containing 1,439 URLs and their context. All the resources of this paper are available here.

## 2. LLMs Exploration for Software and Dataset Mentions Extraction

This section describes the environment for our analyses, the selected LLMs, the used data, and the parsing of the LLMs' output.

### 2.1. Dataset and Software Mentions Extraction Tasks

We investigate the performance of LLMs on two tasks:

**Extraction and Classification (E+CL) Task.** The LLM is prompted to extract URLs and classify them into one of the classes described in Section 2.3. For this task, two prompts are used. *Prompt-1* describes the task and provides four static examples (i.e., provides the same examples for each request to the LLM), and *Prompt-2* describes the task and provides four dynamic examples (i.e., the provided examples are selected based on their textual similarity to the context passed to the LLM).

**Classification (CL) Task** The LLM is provided with the URL and its context and must classify them into one of the predefined classes. For this task, two prompts are used. *Prompt-3* describes the task and provides four static examples, and *Prompt-4* describes the task and provides four dynamic examples.

For both tasks, the LLM is not asked to annotate the provided examples (static and dynamic ones), i.e., the examples are not used in the evaluation. All four prompt templates can be found here and we demonstrate one prompt template for E+CL task in Figure 1.

### 2.2. Large Language Models

In this section, we describe the LLMs used in our exploration:

**LLaMA 2**. LLaMA [17], which stands for "Large Language Model for AI", represents a series of foundational models developed by Meta AI. Launched in two iterations, LLaMA-1 (2022) and LLaMA-2 (2023), it offers various model sizes (7B to 70B parameters) trained on publicly available datasets. Notably, LLaMA-2 surpassed GPT-3 on certain benchmarks while promoting open access with model weights and free use for research and commercial applications.

**Mistral 7B** [18] is an LLM released in 2023. It exploits grouped-query attention (GQA) [19] and sliding window attention (SWA) [20, 21] to speed up the inference and reduce memory requirements during decoding, facilitating the accommodation of larger batch sizes. Furthermore, SWA enables the management of longer sequences, a common problem of several LLMs. These LLMs are suitable to handle long prompts needed to instruct the model to perform the required tasks.

**LLaMA 2 with quantization & Mistral 7B with quantization.** Running the original LLaMA 2 and Mistral models on a GPU demands a relevant amount of vRAM. Since these requirements currently exceed the hardware available on many research computers, quantization has been applied to reduce the models' 16-bit floating-point weights to values between 2 and 8 bits. This modification enables low-setting machines to run these models while accepting small quality losses in the expected outcomes.

**Figure 1:** Prompt template for extraction and classification (E+CL) task. We provide four examples for each few-shot prompting query. In the prompt template, we showcase an example input and output in `Example 1`.

```
[INST]<<sys>>You act as a human annotator. First read the instructions and given examples, then
    only annotate the last given input accordingly without extra words. Your annotation has to use
    valid JSON syntax.<</sys>>

Annotate the URLs in the input and classify the URLs with the following labels:
1. DatasetDirectLink - the URL is for downloading dataset files
2. DatasetLandingPage - the URL is an introduction or a landing page for some dataset entity
3. Software - when the URL is for some software entity
4. Other - the URL does not fall into the above cases

# formatting
Input: text containing one or more URLs.
Output: for each URL span, first output the URL span, then output one of the four above labels.

# Examples:
# Example 1:
Input: Gowalla https://snap.stanford.edu/data/loc-gowalla.html : the pre-processed data that we
    used in the paper can be downloaded here http://dawenl.github.io/data/gowalla_pro.zip .
Output: [{"URL": "https://snap.stanford.edu/data/loc-gowalla.html", "label": "dataset_landing_page
    "},{"URL": "http://dawenl.github.io/data/gowalla_pro.zip", "label": "dataset_direct_link"}]

# Example 2:
Input: {{{ EXAMPLE TEXT }}}
Output: {{{ EXAMPLE OUTPUT JSON }}}

# Example 3:
...

# Example 4:
...
[/INST]
[INST]
# to annotate
Input: {{{INSERT INPUT HERE}}}[/INST]
```

## 2.3. Gold Standard Data

For our study, we use GitHub URLs extracted from the *unarXiv* [22, 23] dataset as initial seed. The data and the source code utilized for the extraction process are available here. For each repository, its *README* file is parsed to retrieve all outgoing URLs and their surrounding contexts. The selected sample used to investigate the LLMs' performance on the mentioned tasks covers 811 GitHub repositories containing a total of 1,439 URLs in their *README* files. Each entry is then manually labeled as one of the following classes:

- **Dataset Direct Link**. This class includes URLs that point to a dataset file (e.g., JSON, CSV or TXT files with tabular data containing labels and/or floating-point numbers), archives

(e.g., ".zip" or ".tar.gz") containing dataset files, or machine learning models (e.g., ".gguf").

- **Dataset Landing Page**. This class contains URLs pointing to an index page or directory that allows the download of one or more dataset files, a software repository that contains source code generating the dataset or downloading it from an external source (e.g. Google Drive), or the URL points to a file in a GitHub repository containing a dataset in some other file in the same repository.
- **Software**. This class includes URLs that point to software snippets, notebooks, source code repositories, etc.
- **Other**. This class includes all the URLs that could not be mapped as *Dataset Direct Link*, *Dataset Landing Page*, or *Software*.

All URLs are manually annotated by researchers from the computer science domain. During the annotation, it is asked to open the URL in a web browser and to assign only one class. The final annotation distribution for the classes is: 120 *Dataset Direct Link*, 678 *Dataset Landing Page*, 355 *Software*, 286 *Other*.

## 2.4. Output Parsing

LLMs take our prompts containing instructions as textual input and generate the most probable response according to the models' parameters. These create outputs that are not structured and, consequently, cannot be automatically elaborated on. This might make it unfeasible to apply LLMs on a large scale. Thus, to enable and facilitate automatic evaluation of LLMs' generated outputs, the following post-processing steps are applied:

**Remove additional conversational opening phrases.** Although the LLMs are instructed to reply using a JSON object without extra words as the output format, they often include opening phrases that do not add any value to the required tasks (e.g., "*I hope this helps!*"). Therefore, we apply a set of text replacement rules to all generated outputs, filtering out these opening phrases.

**Consolidate and convert the structured format string to JSON.** Due to the generative nature, the prompt output string may not present in the requested structured format or may be incomplete (e.g., instead of returning one JSON array containing JSON objects, it might generate a newline-separated enumeration of JSON objects). Thus, we consolidate the LLM's response as much as possible and then convert the JSON fragments into our designed structured format.

**URL Matching.** Multiple URLs can appear in the input context presented to the LLMs. To map the detected URLs with the ground-truth URLs, a 1-to-1 bipartite graph matching solution is used. More precisely, for each ground-truth URL the most similar unmatched detected URL is selected. The similarity is based on the longest common substring ratio against the ground-truth URL. Spurious URLs are matched with the empty set. Despite our effort to extract as much information from the LLMs' outputs as possible, it is not always possible to parse them into an appropriate format for evaluation. We report these parsing statistics for different settings in the evaluation section.

**Table 1**
LLM output string parsing statistics

| Task Setting | Model | Static or Dynamic | Parsed Ratio (%) |
|---|---|---|---|
| E+CL | Llama 2 7b | static | 85.8 |
| E+CL | Llama 2 7b | dynamic | 87.9 |
| CL | Llama 2 7b | static | 92.5 |
| CL | Llama 2 7b | dynamic | 95.4 |
| E+CL | Llama 2 7b qt4bit | static | 95.2 |
| E+CL | Llama 2 7b qt4bit | dynamic | 95.7 |
| CL | Llama 2 7b qt4bit | static | 86.3 |
| CL | Llama 2 7b qt4bit | dynamic | 91.3 |
| E+CL | Mistral 7b | static | 96.9 |
| E+CL | Mistral 7b | dynamic | 96.5 |
| CL | Mistral 7b | static | 96.3 |
| CL | Mistral 7b | dynamic | 96.6 |
| E+CL | Mistral 7b qt4bit | static | 96.1 |
| E+CL | Mistral 7b qt4bit | dynamic | 96.5 |
| CL | Mistral 7b qt4bit | static | 93.3 |
| CL | Mistral 7b qt4bit | dynamic | 92.2 |

## 2.5. Evaluation Setting

We perform the evaluation of the LLMs for the tasks described in Section 2.1 using precision and recall as defined in [24]. More precisely, we compute these scores following this schema:

- **Strict.** The LLM's prediction precisely matches the gold standard annotation in both boundary surface string and entity type.
- **Exact.** The LLM's prediction exactly matches the boundary surface URL of the gold standard annotation, regardless of the entity type.
- **Partial.** The LLM's prediction partially overlaps with the boundary surface URL of the gold standard annotation, regardless of the entity type.
- **Type.** The LLM's predicted entity types are matched with the gold standard annotation, even if the boundary surface string does not fully match.

The different evaluation categories capture various levels of correctness in LLMs' predictions. Furthermore, we also analyze a binary setting where the LLMs' output is only distinguished between those URLs which refer to datasets (labeled as *Dataset Direct Link* or *Dataset Landing Page*) and those which do not. This simpler task can serve as a baseline and help to better analyze the potential trade-off between complexity and accuracy.

## 3. Results and Discussion

This section describes the evaluation we conducted to understand how LLMs can be leveraged to identify dataset and software mentions from GitHub.

**Evaluation on proper output generation.** Table 1 reports statistics regarding the number of generated outputs from the models that correctly adhered to the required input prompt for

**Table 2**

Performance of different LLMs for the E+CL task using FSL. The metrics P and R represent precision and recall separately; *b* refers to the binary results (whether the URLs are related to datasets). The best results for each metric in a specific setting are highlighted in bold and italic.

| Model | Evaluation Setting | Static Few-shot | | | | Dynamic Few-shot | | | |
|-------|--------------------|-------|-------|-------|-------|-------|-------|-------|-------|
| | | **P** | **R** | **P(b)** | **R(b)** | **P** | **R** | **P(b)** | **R(b)** |
| Llama 2 7b | Strict | 0.371 | 0.480 | 0.472 | 0.617 | 0.244 | 0.311 | 0.481 | 0.615 |
| | Exact | 0.711 | *0.918* | 0.703 | *0.918* | 0.703 | 0.894 | 0.700 | 0.894 |
| | Partial | 0.716 | *0.925* | 0.708 | *0.925* | 0.706 | 0.898 | 0.703 | 0.898 |
| | Type | 0.389 | 0.502 | 0.499 | 0.653 | 0.258 | 0.328 | 0.510 | 0.651 |
| Llama 2 7b qt4bit | Strict | 0.354 | 0.371 | 0.572 | 0.609 | 0.319 | 0.367 | 0.568 | 0.670 |
| | Exact | 0.796 | 0.832 | 0.793 | 0.832 | 0.751 | 0.865 | 0.746 | 0.880 |
| | Partial | 0.808 | 0.845 | 0.805 | 0.845 | 0.758 | 0.872 | 0.756 | 0.892 |
| | Type | 0.382 | 0.400 | 0.629 | 0.660 | 0.342 | 0.394 | 0.616 | 0.727 |
| Mistral 7b | Strict | 0.498 | 0.471 | 0.691 | 0.653 | *0.519* | *0.515* | *0.701* | *0.695* |
| | Exact | *0.881* | 0.832 | *0.881* | 0.832 | 0.874 | 0.867 | 0.874 | 0.867 |
| | Partial | *0.884* | 0.835 | *0.884* | 0.835 | 0.876 | 0.869 | 0.876 | 0.869 |
| | Type | 0.512 | 0.483 | 0.734 | 0.694 | *0.537* | *0.532* | *0.749* | *0.743* |
| Mistral 7b qt4bit | Strict | 0.443 | 0.431 | 0.653 | 0.635 | 0.422 | 0.414 | 0.634 | 0.621 |
| | Exact | 0.834 | 0.811 | 0.835 | 0.811 | 0.805 | 0.790 | 0.806 | 0.790 |
| | Partial | 0.839 | 0.815 | 0.839 | 0.816 | 0.813 | 0.797 | 0.814 | 0.797 |
| | Type | 0.472 | 0.459 | 0.698 | 0.679 | 0.448 | 0.439 | 0.687 | 0.673 |

parsing. The percentage of generated outputs that were not analyzable ranges between 3.1% to 14.2%. This issue was particularly prevalent for the Llama models across both static and dynamic settings, whereas Mistral demonstrated greater stability in producing the expected outcomes. This discrepancy significantly impacted subsequent analyses, as outputs deviating from the expected format were considered invalid, negatively affecting the overall models' performance.

**Evaluation on the E+CL Task.** The performance of the LLMs for the E+CL Tasks can be observed in Table 2. Due to a low ratio (e.g., 18/733 for Llama 2 7b) of parsable outputs in ZSL setting, which leads to near-zero precision and recall scores, we only report the FSL results. The LLMs can detect URLs based on the *exact* and *partial* results. However, their success rate is lower than that of known non-LLM-based alternative methods (e.g., regular expressions). This can be explained by our observation that sometimes one or more of the URLs contained in the input context are missed by the LLMs, or nonexisting URLs are hallucinated and appended to the generated output. Among the four LLMs in Table 2, we observe that Mistral 7b outperforms the other models in *strict* and *type* settings, which corresponds to the best classification capability regardless of taking the URL extraction boundary into account. A generally reduced performance of the quantized models is visible when compared with the respective original models, except for the *dynamic* setting of Llama 2 7b. Also, the quantized Mistral model surpasses Llama 2 7b original in the classification task. Furthermore, in contrast with recent literature [25], our experiments do not show an improvement in the models' performance when prompted with dynamic samples. This could indicate that examples need to be carefully selected when feeding

**Table 3**
Performance of different LLMs for the CL task using FSL.

| Model | Evaluation Setting | Static Few-shot | | | | Dynamic Few-shot | | | |
|---|---|---|---|---|---|---|---|---|---|
| | | P | R | P(b) | R(b) | P | R | P(b) | R(b) |
| Llama 2 7b | Strict | 0.394 | 0.436 | 0.611 | 0.677 | 0.240 | 0.256 | 0.576 | 0.615 |
| | Exact | 0.877 | 0.972 | 0.877 | 0.972 | 0.885 | 0.945 | 0.885 | 0.945 |
| | Partial | 0.878 | 0.972 | 0.878 | 0.972 | 0.887 | 0.946 | 0.887 | 0.946 |
| | Type | 0.405 | 0.448 | 0.627 | 0.695 | 0.250 | 0.267 | 0.597 | 0.637 |
| Llama 2 7b qt4bit | Strict | 0.295 | 0.360 | 0.519 | 0.629 | 0.306 | 0.377 | 0.535 | 0.663 |
| | Exact | 0.697 | 0.852 | 0.794 | 0.852 | 0.706 | 0.871 | 0.703 | 0.872 |
| | Partial | 0.708 | 0.866 | 0.715 | 0.866 | 0.717 | 0.884 | 0.714 | 0.886 |
| | Type | 0.325 | 0.397 | 0.580 | 0.702 | 0.329 | 0.406 | 0.589 | 0.730 |
| Mistral 7b | Strict | 0.459 | 0.495 | 0.678 | 0.730 | *0.507* | *0.538* | *0.713* | *0.757* |
| | Exact | 0.864 | 0.930 | 0.864 | 0.930 | *0.894* | *0.949* | *0.894* | *0.949* |
| | Partial | 0.864 | 0.930 | 0.864 | 0.930 | *0.894* | *0.949* | *0.894* | *0.949* |
| | Type | 0.472 | 0.508 | 0.705 | 0.759 | *0.519* | *0.551* | *0.751* | *0.797* |
| Mistral 7b qt4bit | Strict | 0.434 | 0.458 | 0.646 | 0.666 | 0.419 | 0.457 | 0.626 | 0.665 |
| | Exact | 0.781 | 0.823 | 0.801 | 0.826 | 0.770 | 0.841 | 0.792 | 0.842 |
| | Partial | 0.791 | 0.835 | 0.801 | 0.826 | 0.778 | 0.849 | 0.799 | 0.850 |
| | Type | 0.473 | 0.499 | 0.698 | 0.720 | 0.454 | 0.496 | 0.683 | 0.726 |

the models to achieve optimal model performance.

**Evaluation on the CL Task.** The results of the CL Tasks are reported in Table 3. This task should be relatively straightforward compared to the E+CL tasks, given that the URL to be classified was provided as part of the input. Here, in terms of the E+CL task, one of our findings is that the LLMs are unable to perform the task with reasonable precision. Furthermore, our investigation reveals a notable challenge: the models frequently struggled to accurately match the input URL with the URL provided in the context. This results in mismatches between the two URLs or even the generation of entirely new URLs, thereby reducing the effectiveness of the richer input context.

## 3.1. Findings and Limitations

The experience in testing LLMs in ZSL and FSL for the E+CL and CL tasks can be summarized as follows:

**Limited parsing capabilities.** LLMs, due to their generative nature, do not appear suitable to perform tasks requiring parsing and extraction of complex entities such as the ones subject of this study. Although LLMs generally can detect URLs, they lack the precision offered by other non-LLM-based methods (e.g., regular expression based heuristics) and sometimes hallucinate non-existing URLs. Additionally, they struggle to classify URLs into similar but nonidentical classes (in the case of distinguishing *dataset direct link* and *dataset landing page*), thus making their use for certain KG population tasks unfeasible.

**Understanding issue.** Sometimes, the models do not understand the request and provide replies which are not useful or irrelevant for the entity detection task. Examples are "*Sure! I'm*

*ready to annotate the URLs in the input. Please provide the input text.*" and "*Of course, I can't predict the future, and I don't know what will happen to me or to the world.*".

**More input same results limitation.** The fact that a richer input in the CL task did not help the model to perform better highlights a significant limitation in the models' ability to properly integrate and leverage contextual information for niche challenges, emphasizing the need for further research to enhance their contextual understanding and performance in such tasks.

## 4. Conclusion and Outlook

In this paper, we investigated two LLMs and their quantized models, that are open to the scientific community free of charge, for the niche task of aiming at organizing datasets and software into a KG. Despite the considerable enthusiasm surrounding LLMs, our investigation reveals a sobering reality: off-the-shelf models are inadequate for addressing intricate tasks demanding high precision and recall, particularly in the identification and classification of URLs. Effective organization following semantic web best practices necessitates a level of precision and recall that these models fail to achieve. Our findings emphasize the importance of tempering expectations regarding the applicability of LLMs to complex tasks and highlight the need for further research and development to enhance their suitability for such endeavors.

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
