# OpenReview forum: "Hidden Entity Detection from GitHub Leveraging Large Language Models"
_KDD.org/2024/Workshop/DL4KG — DL4KG 2024_

### Official Review · Reviewer_yGed · 2024-07-02
**Interesting work on LLMs for link extraction and classification; needs a baseline**

**Rating:** 6
**Confidence:** 4

**Review:**

This paper presents a study of LLMs applied to link extraction and classification of data extracted from GitHub repositories.

The motivations are clearly explained and the work is rather well organized. The only major flaw is the absence of a non-LLM baseline which could help understand the task's difficulty and therefore give more strength to the conclusions. Given the proposed categories, I think RegEx (Regular Expressions) seems to be a good fit for a baseline.

I'd suggest the authors to highlight the important results in the result tables, to improve readability.

"they often include opening phrases that do not add any value to the required tasks (e.g., “I hope this helps!”)." -> Couldn't this be included in the prompt (eg "do not produce additional phrases, only the output")?

"The LLM is prompted to extract URLs and classify them into one of the classes described in Section 2.5." -> it should be Section 2.3

---

### Official Review · Reviewer_BaY7 · 2024-07-02
**Accept: The paper is well written but lack of significance.**

**Rating:** 7
**Confidence:** 4

**Review:**

This work explores different prompts with Large Language Models (LLMs)  to identify datasets and software mentions within repository texts.

## Pros:

Originality: The authors have built a manually annotated dataset for URL extraction and classification.

Quality: The authors use open LLMs and provide the datasets and the source code.

Clarity: The paper is easy to read.

## Cons:

Significance: Missing comparison with non-LLM -based approaches. The authors mention that “...we present our analysis in FSL settings due to the lack of available training data”, but they do have annotated their gold standard dataset, which could be used as training data.


## Minor remarks:

(1) Abstract: “...The study explores different ZSL prompt learning approaches…” => remove ZSL or change it to FSL

(2) Missing references to related works in Results and Discussion:

“However their success rate is lower than known non -LLM-based alternative methods.”

“Furthermore, in contrast with recent literature, our experiments do not show an improvement in the models’ performance when prompted with dynamic samples.”

## Questions:

(1) Why are the LLMs  instructed to reply using a JSON object as output format ?

(2) “This task diverges from traditional (NER) approaches for three main reasons…” =>  Why LLMs are better than traditional(NER) approaches ?

---

### Official Review · Reviewer_7mDj · 2024-07-08
**Review for "Hidden Entity Detection from GitHub Leveraging Large Language Models"**

**Rating:** 6
**Confidence:** 4

**Review:**

The paper touches upon an interesting problem and provides an LLM-based solution with the help of zero-shot and few-shot learning given that the task is newly introduced and it lacks training data. The paper performs thorough experimentation and the source code is openly accessible.

- Given the general strength of the targeted problem.  The writing of the paper could be strengthened. The introduction suddenly switches from the NER to talking about the repository containing the codes.
- The problem should be introduced with the help of an example.
- Example prompts should be added to the paper since this is the main contribution. Post-processing may not be required to be detailed. I had to open the repository to look at the prompts.
- Why not use LLMs to generate the training dataset?
- How is it important to identify the source code by URL, what is the impact and where can it be used?
- Why is there the word "hidden" in the title of the paper?
- The abstract could be improved with some more concrete insight into the methodology targeted in the paper as well as some results.

---

### Decision · Program_Chairs · 2024-07-09

Accept